# A Hybrid Forecasting Model for Self-Similar Traffic in LEO Mega-Constellation Networks

**Chi Han** [1,2] , **Wei Xiong** [1,2,*] **and Ronghuan Yu** [1,2]

1    Science and Technology on Complex Electronic System Simulation Laboratory, Space Engineering University, Beijing 101400, China; 15850466132@163.com (C.H.); yrh1983@163.com (R.Y.)
2    School of Space Information, Space Engineering University, Beijing 101400, China
\*    Correspondence: 13331094335@163.com

**Abstract:** Mega-constellation network traffic forecasting provides key information for routing and resource allocation, which is of great significance to the performance of satellite networks. However, due to the self-similarity and long-range dependence (LRD) of mega-constellation network traffic, traditional linear/non-linear forecasting models cannot achieve sufficient forecasting accuracy. In order to resolve this problem, a mega-constellation network traffic forecasting model based on EMD (empirical mode decomposition)-ARIMA (autoregressive integrated moving average) and IGWO (improved grey wolf optimizer) optimized BPNN (back-propagation neural network) is proposed in this paper, which makes comprehensive utilization of linear model ARIMA, non-linear model BPNN and optimization algorithm IGWO. With the enhancement of the global optimization capability of a BPNN, the proposed hybrid model can fully realize the potential of mining linear and non-linear laws of mega-constellation network traffic, hence improving the forecasting accuracy. This paper utilizes an ON/OFF model to generate historical self-similar traffic to forecast. RMSE (root mean square error), MAE (mean absolute error), R-square and MAPE (mean absolute percentage error) are adopted as evaluation indexes for the forecasting effect. Comprehensive experimental results show that the proposed method outperforms traditional constellation network traffic forecasting schemes, with several improvements in forecasting accuracy and efficiency.

**Keywords:** satellite network traffic forecast; empirical mode decomposition; autoregressive integrated moving average; BP neural network; adaptive order optimization operator

## 1. Introduction

LEO mega-constellation networks demonstrate excellent communication capabilities, including wide geographic coverage, low delay, large bandwidth and uninterrupted service provision, highlighting their significant role in space–air–ground integrated networks (SAGINs) [1]. In recent years, with the construction of LEO mega-constellations such as Starlink, OneWeb and Kuiper [2], the explosive growth of Internet-connected devices and traffic demand have increased drastically. However, on-board computing and storage resources are limited by power consumption and the size of satellites. Thus, LEO mega-constellations face challenges in capacity and quality of service (QoS) [3]. Due to continuous inter-satellite links (ISLs) switching between adjacent satellites, the network topology changes drastically over time. Meanwhile, satellite network traffic shows temporal and spatial heterogeneity, i.e., traffic demands may vary greatly in different geographical locations or in different time slots [4]. In the case of expanding user scale and traffic demand, the above challenges result in increased congestion in LEO mega-constellations.

Through traffic forecasting, the change in network traffic can be analyzed in advance. The control of network traffic can be changed from reactive response to proactive perception. On the one hand, by forecasting the traffic demand for the next time slot, traffic transmission paths can be planned in advance to proactively avoid congestion. On the other

hand, for bandwidth-sensitive services, always reserving peak bandwidth will inevitably reduce resource utilization. On the basis of traffic forecasting, bandwidth allocation can be dynamically adjusted according to service demand to achieve the efficient utilization of bandwidth resources.

The priority of satellite network traffic forecasting is to build network traffic models that represent the network properties accurately. Traffic demand at different geographic locations is usually cyclical [5]. Meanwhile, the self-similarity of traffic is prevalent in terrestrial networks [6]. In a satellite network, the traffic is aggregated through terrestrial gateways, which will not change the basic properties, i.e., long-range dependence (LRD) on a large scale and self-similarity on a small scale [7], as shown in Figure 1. The actual traffic demand can be divided into long-range baseline and short-range fluctuations; baseline traffic has LRD properties, whereas fluctuations are short-range dependent. Therefore, forecasting models with high precision and low complexity that can demonstrate self-similarity are essential for network traffic forecasting. Traditional network traffic forecasting models, i.e., Poisson models and Markov models, can only deal with short-range dependence (SRD) traffic. Meanwhile, due to the limited on-board computing resources of satellite networks, the computational complexity of a satellite network's traffic forecasting model needs to be reduced [8]. So, the traffic prediction models of terrestrial networks are not suitable for satellite networks with LRD properties.

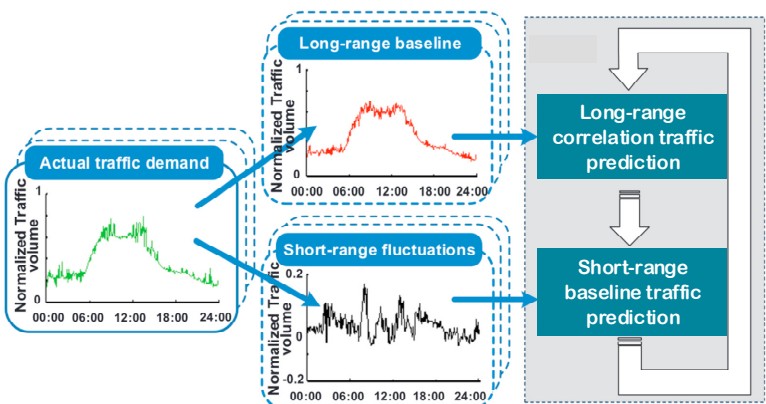

**Figure 1.** Long−range baseline and short−range fluctuations of mega−constellation network traffic.

In order to establish a high-precision and low-complexity traffic forecasting model, it is necessary to first decompose network traffic to extract the long-range correlation traffic and short-range fluctuation traffic. Moreover, the computational complexity of the forecasting models for LRD traffic in terrestrial networks is considerable. Due to the limited on-board computing and storage resources of LEO satellites, it is necessary to reduce the computational complexity. Therefore, the forecasting models of terrestrial networks cannot be adopted to constellation networks directly. It is necessary to establish a high-precision and low-complexity forecasting model for mega-constellation network traffic.

To address the above issues, constellation network traffic is decomposed into multi-order intrinsic mode functions (IMFs) with empirical mode decomposition (EMD). Given the high accuracy of short-term forecasts, the differential integrated moving average autoregressive model (ARIMA) is utilized to forecast IMFs [9]. A back-propagation neural network (BPNN), as a traditional neural network model, has better non-linear fitting ability and performs well in terrestrial network traffic forecasting [10]. BPNNs are utilized to forecast the residual from the ARIMA. However, numerous hyperparameters in BPNNs result in several problems, i.e., slow convergence and local optima [11]. The grey wolf optimization algorithm (GWO) is a group intelligence algorithm derived from the social hierarchy and hunting mechanisms of grey wolves [12]. It has been proved that GWO has a more reasonable global optimum search mechanism, which makes it more suitable for parameter optimization problems. Therefore, GWO is utilized to optimize the hyperparameters

of the BPNN and prevent the BPNN from local optimality. Thus, the self-similar satellite network traffic hybrid forecasting model based on EMD-ARIMA-BPNN is proposed.

In the proposed EMD-ARIMA-BPNN model, constellation network traffic is first decomposed into multi-order IMFs with EMD. Secondly, IMFs are reconstructed to obtain network traffic residuals, which are utilized as the input for a BPNN optimized by GWO. The LRD properties in residuals can be adequately captured by the GWO-BPNN. Finally, the reconstructed IMFs and forecasted residuals are combined as satellite network traffic forecasting results. The main contributions of this paper are described as follows:

- Firstly, the LRD properties of satellite network traffic can result in deteriorated forecasting accuracy and increased computational complexity. Empirical mode decomposition (EMD) is utilized to decompose LRD traffic into multi-order IMFs with single frequencies and SRD to reduce forecasting complexity;
- Secondly, to address the problems of the uneven initial population distribution and slow convergence rate of the standard GWO, an opposition-based learning strategy (OBL) and cosine-based non-linear convergence factor are utilized to improve the GWO, i.e., IGWO. The forecasting accuracy of the BPNN is improved by utilizing IGWO to optimize the hyperparameters;
- Finally, we design the total flow of the LEO mega-constellation traffic forecasting process based on OMNET++. The effectiveness of the proposed satellite network traffic forecasting model is verified through comparative experiments.

The rest of the paper is organized as follows: Section 2 introduces the current research status of satellite network traffic forecasting. Section 3 constructs the satellite network traffic generation model and proposes the EMD-based network traffic decomposition. Section 4 proposes the ARIMA-based IMFs forecasting model and BPNN-based residual forecasting model. The LEO mega-constellation network traffic hybrid forecasting model is constructed. Section 5 summarizes this paper and proposes future work.

## 2. Materials and Methods

Traditional network traffic models are generally derived from Poisson processes, including the Poisson and Markov models [13], which can only describe SRD properties. For satellite network traffic with LRD properties, it is difficult for traditional models to accurately characterize the network. Since the discovery of the self-similarity properties of network traffic in 1994 [14], various self-similarity-based traffic forecasting models have been proposed. One class of models describe the observed traffic by constructing physical models, including ON/OFF models with heavy-tailed distribution [15], $M/G/\infty$ queuing models [7], etc. Another category is statistics-based models, which attempt to simulate network traffic through data fitting. Statistics-based models mainly include linear models and non-linear models. Linear models, i.e., the autoregressive model (AR), moving average model (MA), autoregressive moving average model (ARMA) and autoregressive integrated moving average model (ARIMA) [8], need to set a variety of parameters based on experience. Linear models are only suitable for short-term forecasting and find it difficult to deal with the self-similarity and LRD properties of satellite network traffic.

Due to the limitations of linear models, non-linear models such as the fractional autoregressive integration moving average model (FARIMA), wavelet model and grey model are utilized to forecast network traffic. Zhu et al. [16] proposed the Markov-modulated deterministic model to simulate satellite network traffic, which converts the acquisition, storage and transmission of network traffic into an MMDP/D/1/K queued model. Yan et al. [17] proposed a satellite network traffic forecasting model based on ARMA, which represents current traffic through the weighted summation of multiple historical traffic data. Considering the limited onboard computing and storage capacity, ARMA(2, 1) is adopted to reduce the computational complexity. Chen et al. [18] established a traffic forecasting model with two variables: the geographic longitude of ascending nodes and the time from passing ascending nodes. A prediction method based on a surrogate model is adopted to solve the traffic prediction problem with a time-independent ground traffic distribution. According

to the above analysis, statistics-based forecasting methods have limitations in describing the self-similarity, SRD and LRD laws of satellite network traffic [16].

In view of the limitations of statistics-based models, neural network models are applied to forecast network traffic. Compared with statistics-based models, neural network models outperform them in terms of non-linear mapping capabilities [19]. Zhou et al. [20] proposed a forecasting method, SSA-AWELM, by combing singular spectrum analysis (SSA) and an AdaBoost weighted extreme learning machine (AWELM). In the proposed method, SSA decomposes raw traffic into three parts: trend, periodicity and residual. The AWELM is designed to predict each part separately. Fan et al. [10] combined a recurrent neural network (RNN) and a gated recurrent unit (GRU). The experiment results demonstrated that the proposed RNN-GRU model had advantages in forecasting accuracy. Vinchoff et al. [21] combined a graph convolutional neural network (GCN) and a generative adversarial network (GAN). In GCN-GAN, a GCN is utilized to extract the complex properties of network traffic, whereas a GAN is utilized to model the data structures. Li et al. [22] proposed a forecasting model (LA-ResNet) that utilizes attention mechanism to solve spatiotemporal modeling and predict wireless network traffic. However, the computation complexity of LA-ResNet is too considerable to be applied in LEO mega-constellations.

According to the above analysis, increasing the number of hyperparameters will greatly increase the complexity of neural networks. In order to reduce the computation complexity and improve forecasting accuracy, Cai et al. [23] improved long short-term memory (LSTM) using a genetic algorithm (GA), in which the GA is utilized to optimize the LSTM hyperparameters. Sudhakaran et al. [24] proposed a deep learning approach for cellular traffic forecasting with deep neural networks (DNNs) to model cellular traffic. Traffic volume data are treated as tensors, similar to images, which are then fed to a convolutional neural network. Li et al. [25] proposed the GRU neural network traffic prediction algorithm based on transfer learning. The proposed method combines transfer learning to solve the problem of insufficient online traffic data. The particle filter online training algorithm is adopted to reduce the computation complexity. The above network traffic forecasting methods are comprehensively compared in terms of computation complexity and forecasting accuracy, as shown in Table 1.

**Table 1.** Comparison of network traffic forecasting methods.

| Forecasting Method | Scheme | Computation Complexity | Forecasting Accuracy | Characteristics |
|---|---|---|---|---|
| Statistics-based schemes | ARIMA [1] (2019) | Low | Normal | A variety of parameters need to be set based on experience. |
| | Markov model [16] (2018) | Normal | Normal | Network traffic is converted into an MMDP/D/1/K queued model. |
| | ARMA [17] (2015) | Low | Low | Parameters to be set; ARMA(2,1) to reduce computation complexity. |
| | Surrogate model [18] (2019) | Normal | Normal | Traffic prediction time-independent ground traffic distribution. |
| Neural network schemes | SSA-AWELM [20] (2020) | High | High | Decompose raw traffic into three parts and predict separately. |
| | RNN-GRU [10] (2019) | High | High | Combines RNN and GRU to forecast network traffic. |
| | GCN-GAN [21] (2020) | High | Normal | GCN extracts complex properties; GAN models data structures. |
| | LA-ResNet [22] (2020) | High | High | Attention mechanism. |
| | GA-LSTM [23] (2023) | Normal | Normal | Optimize the hyperparameters of LSTM via GA. |
| | DNN [24] (2020) | High | Normal | Treat traffic volume data as a tensor. |
| | GRU [25] (2021) | Normal | High | Reduce computation complexity via a particle filter online training algorithm. |

To address the above issues, this paper decomposes satellite network traffic with self-similarity into multi-order IMFs with short-range correlation using EMD. The ARIMA, improved using an adaptive fixed-order search operator, is utilized to forecast IMFs with low computation complexity. IGWO is utilized to optimize the hyperparameters of a BPNN, which forecasts the residuals of the EMD-ARIMA. With the enhancement of the global optimization capability of the BPNN, the proposed EMD-ARIMA-BPNN hybrid model can fully realize the potential of mining the linear and non-linear laws of satellite network traffic, hence improving the forecasting accuracy.

## 3. Self-Similarity Analysis of Satellite Network Traffic

### 3.1. LEO Mega-Constellation Networks

The topology of LEO mega-constellation networks can be demonstrated by $G = (V, E)$, where $V$ denotes the satellite nodes and $E$ denotes the inter-satellite links (ISLs) between satellites. The inter-satellite links include intra-plane ISLs and inter-plane ISLs. The former are utilized to connect front and rear satellites in the same orbit, whereas the latter are utilized to connect adjacent satellites in two adjacent orbits [26]. A LEO mega-constellation network integrated with a terrestrial network is shown in Figure 2, including satellites, ground stations, ground terminals, feeder links and ISLs. The onerous communication between ground gateways may lead to congestion of the shortest path during the communication peak [27]. In this case, the giant mesh topology of the LEO mega-constellation network provides multiple candidate paths. Through reasonable load management based on traffic forecasting, high transmission as well as low latency can be taken into account at the same time.

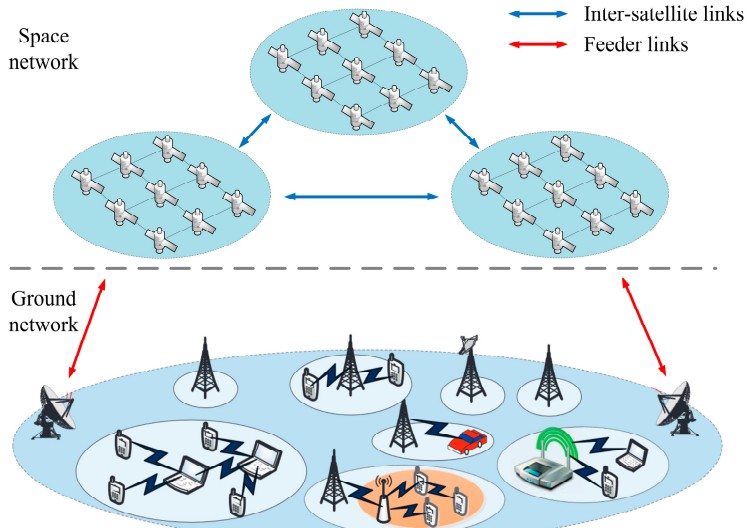

**Figure 2.** Architecture of a LEO mega-constellation network integrated with a terrestrial network.

### 3.2. Traffic Model of a LEO Mega-Constellation Network

In LEO mega-constellation networks, the traffic of sink nodes comes from satellites and ground terminals. Thus, multiple services of multiple nodes converge at the sink nodes of the LEO mega-constellation network. Since traffic transmitted through satellite networks can be described by packet sizes and packet intervals, the source of satellite networks can be regarded as an ON/OFF source that satisfies the Pareto distribution [28].

In this work, the ON/OFF model is utilized to represent the sources of satellite networks alternating between the sending states and non-sending states. The ON state corresponds to the data sending duration, during which sending nodes generate data at a constant rate independently. The OFF state corresponds to pause duration. It can be considered that the ON and OFF duration are independent and identically distributed. According to the above analysis, a LEO mega-constellation network can be regarded as the

joint action of the superposition of numerous ON/OFF sources, as shown in Figure 3. Network traffic is transmitted through the satellite network and finally converges at ground stations, i.e., the receivers in Figure 3.

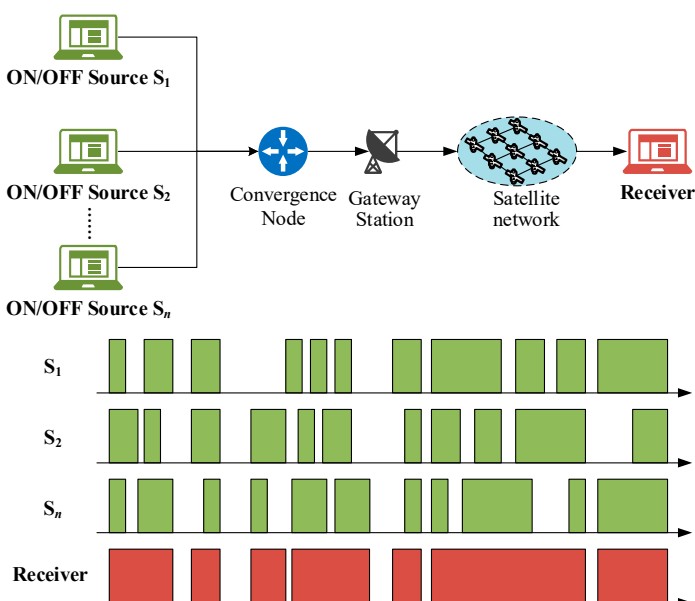

**Figure 3.** Traffic superposition of ON/OFF sources.

Given a source node $S_1$, if the ON duration of $S_1$ satisfies $E(\hat{S}_1) = 1/\mu$, the OFF duration of $S_1$ satisfies $E(\overline{S}_1) = 1/\nu$ and the average speed of the source node is $\mu\nu/(\mu + \nu)$. It has been demonstrated that the traffic generated by the superposition of a series of ON/OFF sources has the property of self-similarity, where the time intervals of the ON and OFF states obey a heavy-tailed distribution [7]. Considering that the Pareto distribution is the most common heavy-tailed distribution, the probability density function and distribution function of Pareto are, respectively,

$$f(x;a,b) = \begin{cases} 0, x \leq b \\ \frac{a}{b}\left(\frac{b}{x}\right)^{a+1}, x > b \end{cases} \tag{1}$$

$$F(x;a,b) = 1 - \left(\frac{b}{x}\right)^{a} \tag{2}$$

where $a$ denotes the shape parameter, which indicates the degree of the heavy tail of the Pareto distribution and $b$ denotes the minimum cut-off parameter, which indicates the lower bound of $x$. Since the ON and OFF durations both obey a heavy-tailed distribution, the superimposition of numerous ON/OFF sources will finally result in the self-similarity of the satellite network traffic.

### 3.3. EMD-Based Satellite Network Traffic Decomposition

Self-similarity results in the long-range dependence (LRD) property of LEO mega-constellation network traffic, i.e., the autocorrelation function of network traffic decreases as a hyperbolic function with increasing time intervals. Although the existing neural-network-based traffic forecasting models are capable of indicating the LRD property, the computation complexity of these models is too considerable to be utilized in LEO mega-constellation networks, where on-board computation and storage resources are extremely limited. Compared with neural network models, statistics-based models outperform them in terms computation complexity, whereas these models can only indicate linear laws of network traffic. The computational complexity can be reduced if the LRD traffic can be

converted into linear components, from which the ARIMA can be adopted to mine the linear laws of the network traffic.

Commonly employed decomposition methods include wavelet [29] and empirical mode decomposition (EMD) [30]. The wavelet's basis is of vital importance to the performance of the wavelet, which limits its application. In contrast, EMD can effectively decompose non-linear and non-stationary data into multi-order IMFs, which can better reflect the inherent characteristics of network traffic [31]. Therefore, EMD is utilized to decompose satellite network traffic in this paper. EMD can decompose LRD traffic into multiple single frequency components and residuals, i.e.,

$$y(t) = \sum_i IMF_i + r(t) \tag{3}$$

where $y(t)$ denotes network traffic, $IMF_i$ denotes the $i$-th components obtained by decomposition and $r(t)$ denotes the residual. Thus, satellite network traffic with LRD properties is converted into several single frequency components, i.e., IMFs, that can be effectively processed by linear models. Meanwhile, the computational complexity is extremely reduced.

The essence of EMD is to determine the basic oscillation pattern of valid signals [32]. Commonly, the IMFs obtained via EMD satisfy the following conditions.

- The number of signal polarization points $\psi$ and the number of zero points $\zeta$ satisfy $|\psi - \zeta| \leq 1$.
- The mean value of the envelope defined by the local maximum and local minimum of the signal is zero.

The first condition specifies that the signal forms of IMFs are consistent with the traditional narrow-band smooth Gaussian process, which can be characterized as

$$y(t) = a(t)e^{j\phi(t)} \tag{4}$$

where $a(t)$ and $\phi(t)$ are the envelope and phase of the signal, respectively. The second condition ensures the symmetry of IMFs. The specific steps of EMD are shown in Table 2.

**Table 2.** EMD processing flow.

| Shifting | IMF$_1$ | IMF$_2$ | IMF$_i$ | IMF$_n$ |
|---|---|---|---|---|
| | $x[n]_{\|L}$ <br> $P = CS\left(x[n]_{\|minimal}\right)_{\|L}$ <br> $Q = CS\left(x[n]_{\|maximal}\right)_{\|L}$ <br> $m_{11}[n] = (P+Q)/2$ | $r_1[n]_{\|L} = x[n] - c_1[n]$ <br> $P = CS\left(r_1[n]_{\|minimal}\right)_{\|L}$ <br> $Q = CS\left(r_1[n]_{\|maxmal}\right)_{\|L}$ <br> $m_{21}[n] = (P+Q)/2$ | ... | $r_{n-1}[n]_{\|L} = r_{n-2}[n] - c_{n-1}[n]$ <br> $P = CS\left(r_{n-1}[n]_{\|minimal}\right)_{\|L}$ <br> $Q = CS\left(r_{n-1}[n]_{\|maxmal}\right)_{\|L}$ <br> $m_{n1}[n] = (P+Q)/2$ |
| 1 | $h_{11}[n] = x[n] - m_{11}[n]$ <br> $if\ \frac{sqr(x[n]-h_{11}[n])}{sqr(x[n])} > 0.3$ <br> then <br> $P = CS\left(h_{11}[n]_{\|minimal}\right)_{\|L}$ <br> $Q = CS\left(h_{11}[n]_{\|maximal}\right)_{\|L}$ <br> $m_{12}[n] = (P+Q)/2$ | $h_{n1}[n] = r_1[n] - m_{n1}[n]$ <br> $if\ \frac{sqr(r_1[n]-h_{21}[n])}{sqr(r_1[n])} > 0.3$ <br> then <br> $P = CS\left(h_{21}[n]_{\|minimal}\right)_{\|L}$ <br> $Q = CS\left(h_{21}[n]_{\|maximal}\right)_{\|L}$ <br> $m_{22}[n] = (P+Q)/2$ | ... | $h_{n1}[n] = r_{n-1}[n] - m_{n1}[n]$ <br> $if\ \frac{sqr(r_{n-1}[n]-h_{n1}[n])}{sqr(r_{n-1}[n])} > 0.3$ <br> then <br> $P = CS\left(h_{n1}[n]_{\|minimal}\right)_{\|L}$ <br> $Q = CS\left(h_{n1}[n]_{\|maximal}\right)_{\|L}$ <br> $m_{n2}[n] = (P+Q)/2$ |
| 2 | $h_{12}[n] = h_{11}[n] - m_{12}[n]$ | $h_{22}[n] = h_{21}[n] - m_{22}[n]$ | ... | $h_{n2}[n] = h_{n1}[n] - m_{n2}[n]$ |
| ... | ... | ... | ... | ... |
| k | $h_{1k}[n] = h_{1(k-1)}[n] - m_{1k}[n]$ <br> else $c_1[n] = h_{1k}[n]$ | $h_{2k}[n] = h_{2(k-1)}[n] - m_{2k}[n]$ <br> else $c_2[n] = h_{2k}[n]$ | ... | $h_{nk}[n] = h_{n(k-1)}[n] - m_{nk}[n]$ <br> else $r_n[n] = h_{nk}[n]$ |

**Step 1:** Given a signal $x(t)$ to be analyzed, all polar points are fitted with two cubic spline curves to obtain the polar envelope of $x(t)$.

**Step 2:** Let the average value of the envelope be $m(t)$, the remaining signal $h(t)$ can be indicated by $h(t) = x(t) - m(t)$. $h(t)$ will be taken as $IMF_1$ if it satisfies the above two conditions. Otherwise, $h(t)$ will be taken to replace $x(t)$ and the above steps will be repeated.

**Step 3:** After $k$ iterations, the difference between the obtained signal and the mean value of the envelope is $h_{1,k}(t)$. The difference obtained in the $k-1$ iteration is indicated as $h_{1,k-1}(t)$. $h_{1,k}(t)$ will be taken as the first IMF when

$$\sqrt{\sum_{t=0}^{T} \frac{[h_{1,k-1}(t) - h_{1,k}(t)]^2}{h_{1,k-1}(t)^2}} < \xi \tag{5}$$

where $\xi$ denotes the threshold value. After $k$ iterations, the root mean square error (RMSE) between $h_{1,k}(t)$ and $h_{1,k-1}(t)$ is less than $\xi$. Then, $h_{1,k}(t)$ will be taken as the first IMF that satisfies the condition.

**Step 4:** Take $l(t) = x(t) - h_{1,k}(t)$ as $x(t)$ and repeat the above steps. When the remaining residual is monotonical and the amplitude is less than $\xi$, several IMFs, $C_i(t)$ and a final residual $r(t)$ are obtained, i.e.,

$$x(t) = \sum_{i=1}^{n} C_i(t) + r(t) \tag{6}$$

**Lemma 1.** *Given the LEO mega-constellation network traffic $D(t)$ with LRD properties. The IMFs of $D(t)$ decomposed using EMD have the property of short-range dependence.*

The proof of Lemma 1 is detailed in Appendix A.

Commonly, the IMFs decomposed via EMD mainly include three categories: noise-dominated, noise mixed with valid information and valid-information-dominated IMFs. The Pearson correlation coefficient, Shannon information entropy and Hurst index are usually utilized to classify IMFs. However, after identification, noise-dominated IMFs tend to be discarded, resulting in the loss of valid information. To concretely illustrate this issue, the valid information contained in $IMF_i (i = 1, 2, \cdots, n)$ is shown in Figure 4.

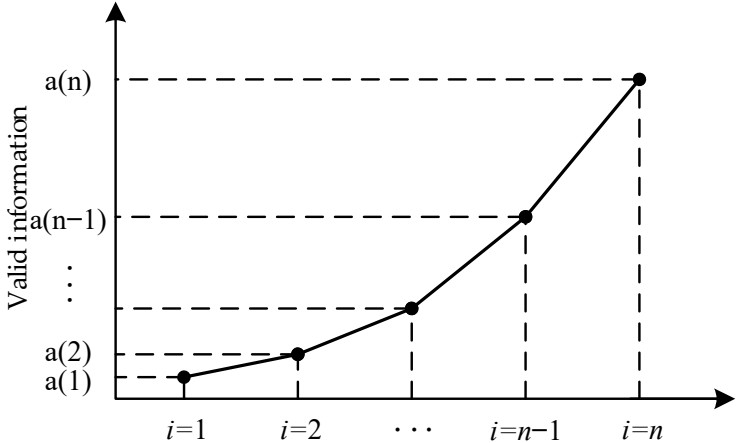

**Figure 4.** Valid information ratio of $IMF_i$ ($i$ = 1, 2, …, $n$).

Let the valid information ratio of $IMF_i$ be as $a(i)$:

$$a(i) = \frac{\Gamma(IMF_i)}{\Gamma(D(t))} \tag{7}$$

where $\Gamma(\text{IMF}_i)$ denotes the valid information of $\text{IMF}_i$ and $\Gamma(D(t))$ denotes the valid information of the raw satellite network traffic $D(t)$. With an increase in order, the valid information ratio increases as well. The valid information ratios of all IMFs satisfy

$$\sum_{i=1}^{n} a(i) = 1 \tag{8}$$

It can be seen that low-order IMFs still contain a small amount of valid information. Thus, discarding low-order IMFs will result in the loss of valid information. Meanwhile, since high-order IMFs contain noise, adopting high-order IMFs directly may inevitably introduce noise. Hence, in contrast to the existing works, the IMFs will not be classified in the EMD proposed in this paper. Instead, all IMFs will be adopted in the ARIMA to obtain residuals and multiple single frequency components.

## 4. Network Traffic Forecasting Based on ARIMA-BPNN

*4.1. ARIMA-Based IMF Forecasting*

4.1.1. ARIMA

To address the problem of the LRD laws of satellite network traffic, EMD is utilized to decompose traffic into multiple IMFs and residuals. The autoregressive integrated moving average model (ARIMA) is a forecasting method proposed by Box and Jenkins in the 1970s [9] that has advantages in short-term forecasting [8]. Thus, the EMD-ARIMA combined model is constructed, where ARIMA is utilized to forecast the IMFs obtained via EMD.

Take IMFs as input, ARIMA($p$, $d$, $q$) is utilized to forecast the IMFs in turn. The ARIMA($p$, $d$, $q$) model is given by

$$\nabla^d X(t) = \sum_{i=1}^{p} \phi_i \nabla^d X(t-i) - \sum_{j=1}^{q} \theta_j \varepsilon(t-j) + \varepsilon(t) \tag{9}$$

$$\text{s.t.} \begin{cases} \phi_p \neq 0, \theta_q \neq 0 \\ E(\varepsilon(t)) = 0, Var(\varepsilon(t)) = \sigma_\varepsilon^2 \\ E(\varepsilon(t)\varepsilon(s)) = 0, s \neq t \\ E(\nabla^d X(s)\varepsilon(s)) = 0, \forall s < t \end{cases} \tag{10}$$

where $\nabla^d X(t)$ denotes the value of the original traffic at time slot $t$ after the differencing of order $d$, $p$ denotes the highest autoregressive order of the differenced series data, $q$ denotes the highest moving-average order of the differenced series data, $\phi_i$ denotes the $i$-th order autoregressive term coefficient, $\theta_j$ denotes the $j$-th order moving-average term coefficient, $E(\varepsilon(t))$ denotes the mean of a random series $\varepsilon(t)$ and $Var(\varepsilon(t))$ denotes the variance of $\varepsilon(t)$, i.e., $\sigma_\varepsilon^2$.

The IMFs of satellite network traffic are converted to a smooth series by the means of difference operation. According to the smooth data decomposition theorem, the network traffic data can be decomposed into deterministic smooth data and a stochastic white noise, which can be indicated equivalently by ARIMA($p$, $d$, $q$).

4.1.2. Order Determination and Parameter Estimation for the ARIMA

Multi-order IMFs are adopted as the input of the ARIMA in this paper. In traditional ARIMA($p$, $d$, $q$), the autocorrelation coefficient (ACF) and partial autocorrelation coefficient (PACF) are commonly utilized to determine the autoregressive order $p$ and moving-average order $q$. However, considering satellite network traffic has the properties of self-similarity and LRD, there is commonly a slight fluctuation near zero, making theoretical truncation difficult to obtain for ACF or PACF. To address this problem, this paper takes the Akaike information criterion (AIC) as the objective function and twice the standard deviation, i.e., $2\sigma_{ACF}$ and $2\sigma_{PACF}$, as the threshold. An adaptive fixed-order optimization

search operator (AFOS) is designed to reduce the interpretation error. The pseudo code of $ARIMA(p, d, q)$ is shown in Algorithm 1.

---

**Algorithm 1:** Order Determination and Parameter Estimation for ARIMA

---

1    **Input:** $IMF_i$, **order of IMF:** $n$
2    **Output:** $ARIMA(p, d, q)$, $\zeta_i(t)$, $IMF_i'(t)$
3    **for** $i = 1; i < n; i + +$ **do**
4        $IMF_{i,d}(t) \leftarrow IMF_i(t), d \leftarrow 0$
5        ADF test for $IMF_{i,d}(t)$
6        $ACF_{i,d}(s), 2\sigma_{ACF}, PACF_{i,d}(s), 2\sigma_{PACF}, s \in \{1,2,3,4,5\}$
7        $s_1 \leftarrow s_{p,\max}$, $s_2 \leftarrow s_{q,\max}$
8        **if** $s_2 = 5$ **then**
9            $q_{\max} \leftarrow 0$
10       **else**
11           $q_{\max} = s_{q,\max} - 1$
12       **end**
13      **if** $s_1 = 5$ **then**
14          $p_{\max} \leftarrow 0$
15      **else**
16          $p_{\max} = s_{p,\max} - 1$
17      **end**
18      $ARIMA(p_i, d_i, q_i), p_i \in \{1, 2, \cdots, p_{\max}\}, q_i \in \{1, 2, \cdots, q_{\max}\}$
19      $\phi_i, i \in \{1, 2, \cdots, p\}; \theta_j, j \in \{1, 2, \cdots, q\}; \sigma_{\varepsilon,p,d,q}^2$
20      $AIC_{p,d,q} \leftarrow T \cdot \ln\left(\sigma_{\varepsilon,p,d,q}^2\right) + 2(p + q + 1)$
21      $ARIMA(p, d, q) \leftarrow ARIMA(p_i, d_i, q_i)|_{\min\{AIC\}}$
22      $IMF_i'(t) \leftarrow d - \text{order differential reduction of } IMF_{i,d}'(t)$
23      $\zeta_i(t) \leftarrow IMF_i'(t) - IMF_i'(t)$
24    **end**

---

The modeling steps of $ARIMA(p, d, q)$ are as follows:

**Step 1:** Set the order of IMF as $I$ and $i = 1$. Set the total number of IMFs as $I$.

**Step 2:** Set the difference order $d = 0$. $IMF_{i,d}(t) = IMF_i(t)$. Take $IMF_{i,d}(t)$ as the input sequence for modeling.

**Step 3:** Perform the ADF test for $IMF_{i,d}(t)$ to determine whether the input sequence is a stationary sequence. If yes, perform step 5, otherwise, perform step 4.

**Step 4:** Perform the difference operation on $IMF_{i,d}(t)$. $d = d + 1$. Assign the result of the difference operation to $IMF_{i,d}(t)$. Perform step 5.

**Step 5:** Perform the white noise test based on LB statistics to determine whether the input noise is white noise. If yes, perform step 17, otherwise, perform step 6.

**Step 6:** Calculate the autocorrelation coefficient $ACF_{i,d}(s)$, corresponding standard deviation $2\sigma_{ACF}$, partial autocorrelation coefficient $PACF_{i,d}(s)$ and corresponding standard deviation $2\sigma_{PACF}$. $s \in \{1, 2, 3, 4, 5\}$.

**Step 7:** $s$ turns from 1 to 5. Judge whether $ACF_{i,d}(s) \geq 2\sigma_{ACF}$ and $PACF_{i,d}(s) \geq 2\sigma_{PACF}$ hold.

**Step 8:** Judge whether $s_{q,\max} = 5$ holds. If yes, $q_{\max} = 0$, otherwise, $q_{\max} = s_{q,\max} - 1$.

**Step 9:** Judge whether $s_{p,\max} = 5$ holds. If yes, $p_{\max} = 0$, otherwise, $p_{\max} = s_{p,\max} - 1$.

**Step 10:** Permutation and combination $(p, q)$ to construct $ARIMA(p, d, q)$ models.

**Step 11:** Calculate parameters in $ARIMA(p, d, q)$ by maximum likelihood estimation (MLE).

**Step 12:** Calculate the $AIC_{p,d,q}$ of different $ARIMA(p, d, q)$ using the AIC criterion:

$$AIC_{p,d,q} = T \ln\left(\sigma_{\varepsilon,p,d,q}^2\right) + 2(p + q + 1) \tag{11}$$

**Step 13:** Take the $ARIMA(p, d, q)$ with the minimum $AIC_{p,d,q}$ as the optimal model.

**Step 14:** Calculate the fitting sequence $IMF_{i,d}^{Fit}(t)$ of $IMF_{i,d}(t)$ by the optimal model.

**Step 15:** Perform the $d$-order differential reduction of $IMF_{i,d}(t)$ and assign the result of reduction to $IMF_i^{Fit}(t)$.

**Step 16:** Calculate the fitted residual sequence $\xi_i(t) = IMF_i^{Fit}(t) - IMF_i(t)$.

**Step 17:** Judge whether $i < I$. If yes, $i = i + 1$ and perform step 2, otherwise, end the modeling step.

Specifically, the modeling steps of AFOS (Step 5–Step 13) are shown in Figure 5.

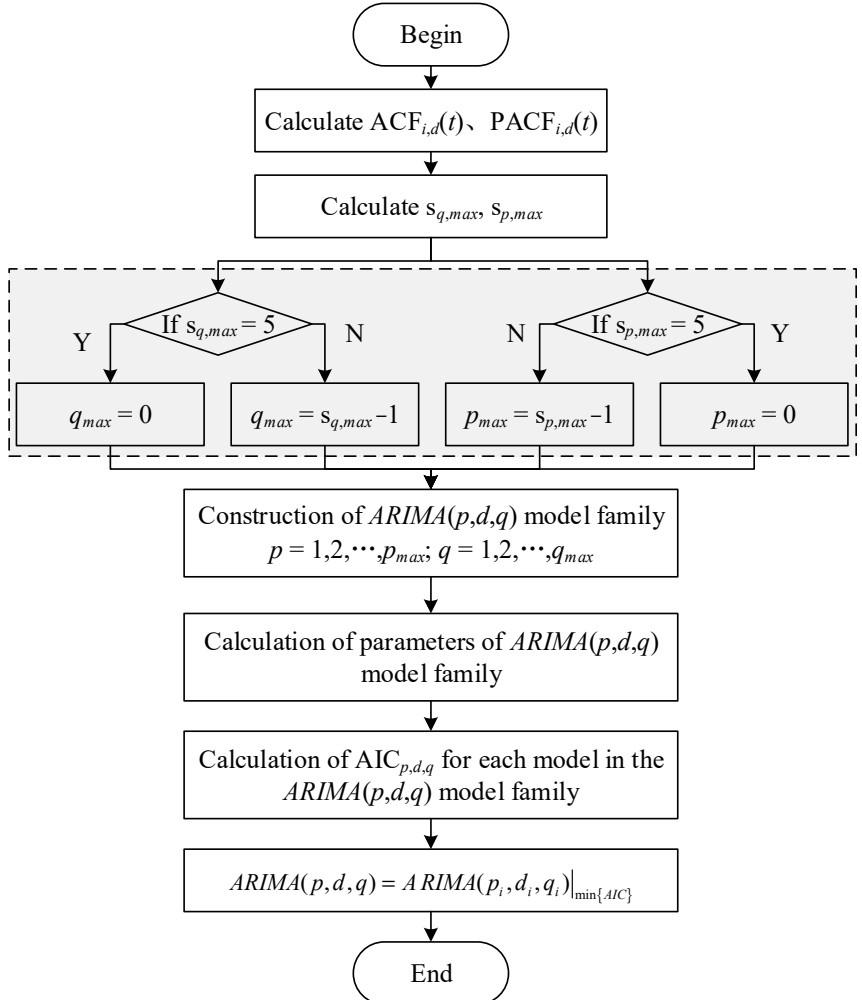

**Figure 5.** Modeling steps of the adaptive fixed−order optimization searching operator.

According to Lemma 1, the IMFs obtained via EMD have an SRD property. Thus, the fixed-order optimization search operator can identify the truncation property of the ACFs. Meanwhile, the $p$ and $q$ of ARIMA($p, d, q$) can also be limited within the third order, which reduces the computational complexity.

### 4.2. IGWO-Based BPNN Parameter Optimization

In the proposed ARIMA-BPNN hybrid model, numerous hyperparameters of BPNN may result in unstable results for residual forecasting. The grey wolf optimization (GWO) algorithm has been proven to have a reasonable global optimum search mechanism for parameter optimization problems [12]. Therefore, GWO is utilized to optimize the hyperparameters of the BPNN in this paper. However, standard GWO has the disadvantages of an uneven initial population distribution and slow converge rate. To address these issues, GWO is improved by adjusted the population initialization strategy and convergence factor to accelerate the convergence. Finally, the improved GWO (i.e., IGWO) is utilized to optimize the hyperparameters of the BPNN.

### 4.2.1. Standard GWO Algorithm

GWO is a group intelligence optimization algorithm derived from simulating the social hierarchy and hunting mechanisms of grey wolves [12]. In GWO, wolves are divided top-down into four groups based on fitness values: $(\alpha, \beta, \delta, \omega)$. The former three groups $(\alpha, \beta, \delta)$ denote the leader class (optimal solution). Candidate solutions are positioned around $(\alpha, \beta, \delta)$ for updating. The position update of $(\alpha, \beta, \delta, \omega)$ consists of three stages.

- Encirclement phase. Based on the hunting encirclement mechanism of grey wolves, the encirclement phase is defined as

$$Q = \left| E X_p(t) - X(t) \right| \tag{12}$$

$$X(t+1) = X_p(t) - LQ \tag{13}$$

where Equation (11) defines the distance between the grey wolves and the prey. Equation (12) defines the position updating of grey wolves. $t$ denotes the number of iterations. $X(t)$ denotes the position of the grey wolves, $X_p(t)$ denotes the position of the prey and $Q$ denotes the distance between the grey wolves. $LE$ is the coefficient vector that is defined as follows

$$L = 2ar_1 - a \tag{14}$$

$$E = 2r_2 \tag{15}$$

$$a = 2 - 2t/T_{\max} \tag{16}$$

where the convergence factor $a$ decreases linearly with the number of iterations within $[2, 0]$, $T_{max}$ is the maximum number of iterations and the variables $r_1, r_2$ are randomly distributed within $[0, 1]$.

- Hunting phase. During the hunting phase, $\alpha, \beta, \delta$ lead the search process and gradually approach the prey. The tracking process can be described as follows

$$\begin{cases} Q_\alpha = |E_1 \cdot X_\alpha - X|, Q_\beta = \left| E_2 \cdot X_\beta - X \right| \\ Q_\delta = |E_3 \cdot X_\delta - X| \end{cases} \tag{17}$$

where $Q_\alpha, Q_\beta, Q_\delta$ denote the distance between $\alpha, \beta, \delta$ and $\omega$, $X_\alpha, X_\beta, X_\delta$ denote the current positions of $\alpha, \beta, \delta$, $X$ is the current position of $\omega$ and $E_1, E_2, E_3$ are random vectors.

$$\begin{cases} X_1 = X_\alpha - L_1 Q_\alpha, X_2 = X_\beta - L_2 Q_\beta \\ X_3 = X_\delta - L_3 Q_\delta \end{cases} \tag{18}$$

$$X(t+1) = \frac{X_1 + X_2 + X_3}{3} \tag{19}$$

where Equation (17) defines the direction of the approximation of $\omega$ to $\alpha, \beta, \delta$ and the corresponding step size. Equation (18) defines the final position of $\omega$.

- Attacking phase. As $(\alpha, \beta, \delta, \omega)$ approach the prey, the convergence factor $a$ decreases linearly and $|L|$ varies within $[-a, a]$. When $|L| < 1$, the algorithm converges and the prey position is obtained.

### 4.2.2. Improvement of the GWO Algorithm

- OBL-Based Population Initialization

In the standard GWO algorithm, $(\alpha, \beta, \delta, \omega)$ are initialized randomly, which tends to result in the uneven distribution of the initial population and affects the convergence performance. In order to enable individuals in the initial population to make the best possible use of the solution space information, we introduce the opposition-based learning (OBL) strategy for population initialization. The essence of OBL is to evaluate the optimal solution obtained from the current search while considering the opposite of the optimal solutions, as shown in Figure 6.

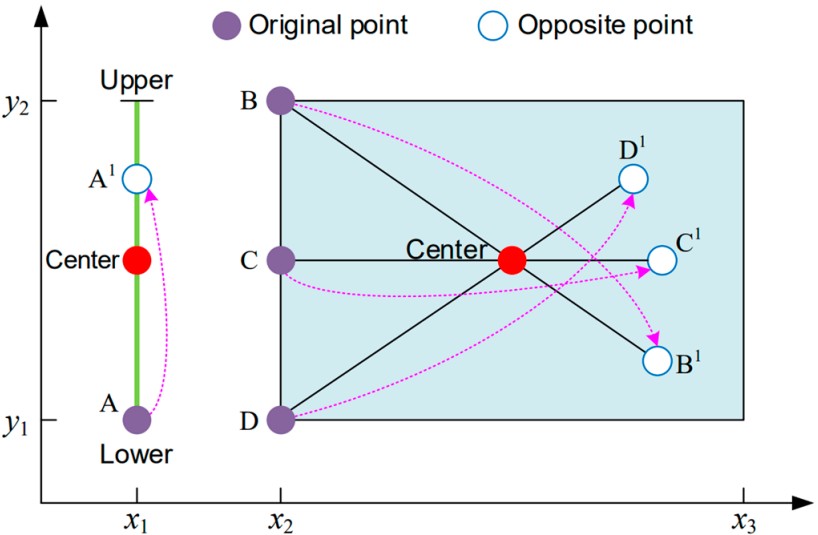

**Figure 6.** Opposition-based learning strategy. B, C and D denote the optimal solutions; the reverse solutions ($B^1$, $C^1$ and $D^1$) are obtained through OBL strategy.

The OBL strategy is specifically defined as follows

$$\widetilde{X}_{ij}^t = \begin{cases} C_j + (C_j - X_{ij}^t) \cdot r_3, \ X_{ij}^t < C_j \\ C_j - (C_j - X_{ij}^t) \cdot r_3, \ X_{ij}^t \geq C_j \end{cases} \tag{20}$$

$$C_j = \frac{\overline{X}_j + \underline{X}_j}{2} \tag{21}$$

where $\widetilde{X}_{ij}^t$ denotes the reverse point of the *i*-th individual in the *j*-th dimension in the *t*-th iteration, $\overline{X}_j \underline{X}_j$ denote the upper and lower boundaries of the *j*-th dimension, respectively, and $r_3$ is randomly distributed within [0, 1].

- Cosine non-linear convergence factor

As a swarm intelligent optimization algorithm, it is difficult for GWO to strike a reasonable balance between a local search and a global search, i.e., it is easy for the standard GWO to converge prematurely. The convergence factors $a \in [0, 2]$ in Equation (13) to Equation (15) linearly decrease, making it difficult to reasonably describe the actual nonlinear convergence process of GWO. In this paper, a cosine non-linear convergence factor is introduced to describe the convergence process, i.e.,

$$a = 2\cos\left(\frac{\pi}{2}\left(\frac{t}{T_{\max}}\right)^2\right) \tag{22}$$

where *t* represents the current iterations and $T_{\max}$ is the maximum number of iterations. The IGWO algorithm is obtained by improving the GWO with the above two strategies.

### 4.2.3. IGWO-Based BPNN Parameter Optimization

In IGWO-BPNN, the hyperparameters of the BPNN are optimized as the position of the grey wolves. By updating the position, the hyperparameters of the BPNN are dynamically optimized, thus obtaining the global optimal solution. The parameter optimization process of the BPNN based on IGWO is shown in Figure 7.

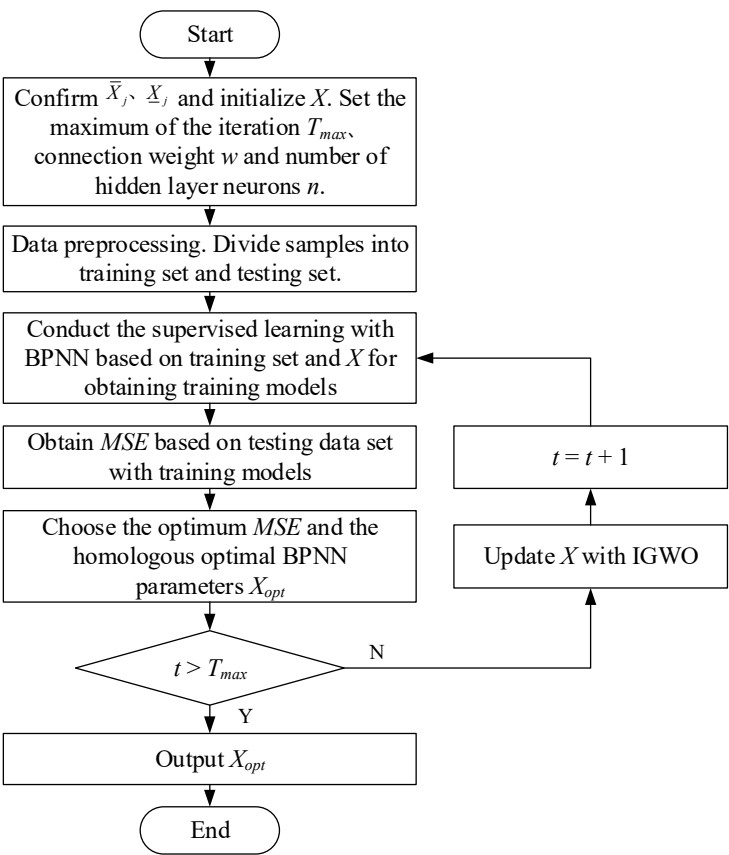

**Figure 7.** Flow of IGWO-based BPNN parameter optimization.

The steps to optimize the hyperparameters of the BPNN through IGWO are as follows:

**Step 1**: Initializing parameters of IGWO and BPNN. The initial position of the grey wolves is determined according to Equations (17) and (18). The population size is set to $M$. The maximum number of iterations is $T_{max}$. The upper and lower boundaries of the $j$-th dimension are set to $\overline{X}_j\underline{X}_j$, respectively. The connection weight of each layer of the neural network is $\omega$. The error threshold is $\varepsilon$, and the number of neurons in the hidden layer is $n$.

**Step 2**: Data preprocessing. In order to eliminate the influence of the residual dimension, the residuals are normalized to $[-1,1]$ according to Equation (22) and divided into a training dataset and a testing dataset.

$$x_i = \frac{2(x'_i - x_{\min})}{x_{\max} - x_{\min}} - 1 \tag{23}$$

where $x_i$ denotes the normalized data, $x'_i$ is the $i$-th raw data and $x_{\max}$ and $x_{\min}$ represent the maximum and minimum of the corresponding data, respectively.

**Step 3**: The termination condition (i.e., the current solution is the minimum) is determined as follows: If a value remains constant in continuous iterations, it is considered to be the minimum value. The selected sample performs **Step 4–Step 7** below until the termination condition is met and then exits.

**Step 4**: The position of each individual contains the hyperparameters of the BPNN, which are trained on the training dataset according to Equation (16) to Equation (18) to constructing the corresponding BPNN model.

**Step 5**: The performance of the BPNN was evaluated with the testing dataset and mean square error (MSE) was utilized to calculate the fitness $fit_i$.

**Step 6**: The grey wolf population was graded according to $fit_i$. The best adapted individual was retained, and the remaining individuals were updated according to Equation (16) to Equation (18).

**Step 7**: If the number of iterations reaches $T_{\max}$, the hyperparameter optimization will end and the loop will jump out. The output of the individual position $(\varepsilon_i, n_i)$ with the best $fit_i$ is selected as the optimal hyperparameters of the BPNN to construct the prediction model.

*4.3. Design of the EMD-ARIMA-BPNN Hybrid Traffic Forecasting Model*

Through the above analysis, the satellite network traffic described by the heavy-tailed ON/OFF superposition model has both linear and non-linear characteristics. ARIMA has higher accuracy in linear data forecasting, whereas the BPNN has better performance in forecasting non-linear data. In this paper, after decomposing satellite network traffic using EMD, the ARIMA was utilized to extract the linear laws of traffic. Meanwhile, the residual of ARIMA will be utilized as an input for a BPNN to extract the non-linear laws. The overall flow chart of the EMD-ARIMA-BPNN is demonstrated in Figure 8. The concrete steps of the EMD-ARIMA-BPNN hybrid model for satellite network traffic forecasting are as follows.

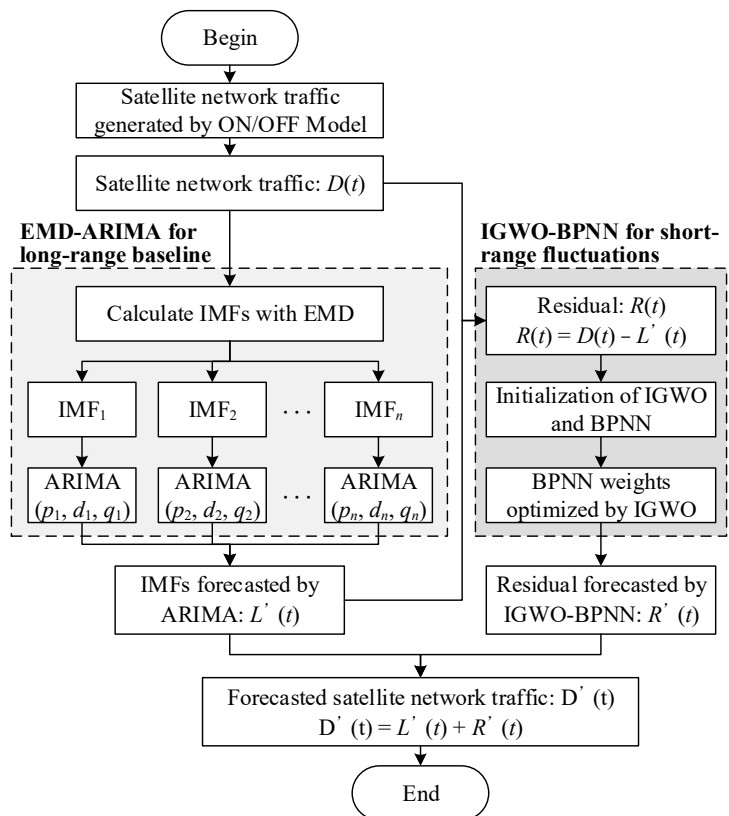

**Figure 8.** EMD-ARIMA-BPNN hybrid model for satellite network traffic forecasting.

**Step 1**: Satellite network traffic generation based on the ON/OFF model. Network traffic is aggregated to the satellite network through gateway stations to generate satellite network traffic data $D(t)$.

**Step 2**: Decompose $D(t)$ into multiple IMFs with short-range dependence using EMD. Batch modelling of multi-order IMFs is implemented according to Algorithm 1 to construct ARIMA$(p_i, d_i, q_i)(i = 1, 2, \cdots, n)$. The final linear sequence $L'(t)$ of the satellite network traffic sequence is obtained.

**Step 3**: Eliminating the linear sequence $L'(t)$ from the raw satellite network traffic $D(t)$ to obtain the residual sequence $R(t)$:

$$R(t) = D(t) - L'(t) \tag{24}$$

**Step 4**: The IGWO algorithm is utilized to optimize the hyperparameters of the BPNN, i.e., $(\varepsilon_i, n_i)$. The residual $R(t)$ is forecasted; $R'(t)$ demonstrates the forecasting result.

**Step 5**: Combining the linear sequence $L'(t)$ of the satellite network traffic obtained from ARIMA$(p_i, d_i, q_i)$ and the forecasted residual $R'(t)$, the forecasted satellite network traffic $D'(t)$ is obtained:

$$D'(t) = L'(t) + R'(t) \tag{25}$$

## 5. Simulation Performance and Analysis

### 5.1. Dataset Description

We construct a LEO mega-constellation with 1584 satellites deployed in 72 orbits on the NS 3.31 simulation platform to evaluate the effectiveness of the proposed method. The simulation platform is deployed on Ubuntu 20.04. Each satellite has four ISLs, including two inter-plane ISLs and two intra-plane ISLs. All ISLs are optical links, which mitigate the influence of interference. The up/downlink rates and bandwidths of the ISLs are both 5 Mbps [33]. The simulation time is 100 s per run. The other parameters of the constellation and network are shown in Table 3.

**Table 3.** Parameter settings of the LEO mega-constellation network.

|  | Parameter | Value |
|---|---|---|
| Orbital parameters | Number of orbits | 72 |
|  | Satellites per orbit | 22 |
|  | Orbital altitude (km) | 550 |
|  | Inclination (○) | 53.8 |
| Network parameters | Buffer queue size | 100 |
|  | Minimum elevation of gateway (○) | 30 |
|  | Uplink/downlink rate (Mbps) | 5 |
|  | Inter-satellite link rate (Mbps) | 5 |
|  | Packet size (byte) | 512 |

According to the satellite network traffic forecasting model in Figure 8, the source satellites alternate between the sending/non-sending states according to the ON/OFF model to generate network traffic. After that, the traffic generated by terrestrial sources will be overlaid to the mega-constellation network through ground stations, obtaining the network traffic of corresponding satellites. The distribution of satellites and ISLs in the simulated constellation is shown in Figure 9.

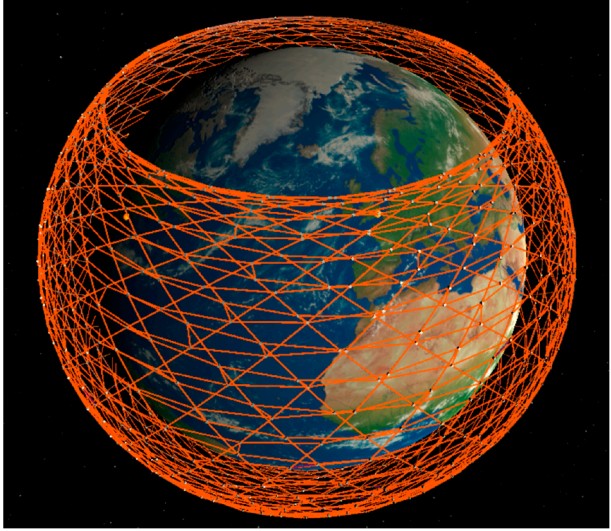

**Figure 9.** Inter-satellite links of a LEO mega-constellation network.

We set eight flows (source and destination pairs) distributed between 40°S and 60°N (most of the global population is distributed within this latitude). The specific sources and destinations of the constellation network traffic are demonstrated in Table 4. The network topology changes over time with the movement of satellites. During the experimental cycle, sampling is performed every minute; $T$ = 1450 traffic data for the whole cycle are selected as the dataset.

**Table 4.** Source and destination of flows.

| Flow Number | Source | Destination |
|:---:|:---:|:---:|
| 1 | Tokyo | Delhi |
| 2 | Shanghai | São Paulo |
| 3 | Mumbai | Mexico City |
| 4 | Beijing | Osaka |
| 5 | Cairo | New York |
| 6 | Dhaka | Karachi |
| 7 | Buenos Aires | Kolkata |
| 8 | Istanbul | London |

*5.2. Self-Similarity Analysis and EMD-Based Decomposition*

This section analyses the self-similarity of the satellite network traffic and decomposes the traffic into short-range dependent traffic using EMD. In this work, self-similarity is measured using the Hurst index ($H$), which is a general index to detect whether a stochastic process is self-similar [34]. $H \in (0, 0.5]$ indicates that network traffic is short-range dependent, whereas $H \in (0.5, 1)$ indicates that network traffic has self-similarity and long-range dependence. A larger $H$-index indicates a higher self-similarity degree, burstiness and dependency. Estimation methods for the Hurst index mainly include residual variance, periodogram, R/S graph and wavelet analysis, among which R/S graph has better robustness for analyzing the edge distribution of random processes. Meanwhile, R/S graph has better adaptability to different distributions. Therefore, R/S graph was adopted to estimate the Hurst index of the satellite network traffic. For the raw traffic demonstrated in Figure 10, $H$ = 0.9414, indicating the strong self-similarity and long-range dependence of the raw satellite network traffic.

The raw traffic is decomposed according to Equation (6) and the steps demonstrated in Table 2. The resulting 10 IMFs (IMF1~IMF10) are shown in Figure 10. It can be found that when the raw traffic is decomposed into 10 IMFs, the magnitude of the residuals is very low and negligible. As the order of IMFs increases, the residuals tend to decrease. Therefore, 10 IMFs are sufficient to meet the forecasting accuracy requirements while reducing the computational complexity of the EMD decomposition.

The envelopes of the autocorrelation function of the raw traffic and IMFs are shown in Figure 11. The grey region is the envelope of the autocorrelation function of the raw traffic, whereas the blue region denotes the superposition of the envelopes of the IMFs autocorrelation functions. It can be seen that the decay of the raw traffic autocorrelation function is similar to a hyperbolic function, with a slow decay rate. After EMD decomposition, the decay rate of the autocorrelation function of the IMFs is significantly faster than that of raw traffic, which indicates that EMD achieves a reduction in self-similarity, making the decomposed components exhibit short-range dependence instead of long-range dependence. Thus, it is proved theoretically and experimentally that the satellite network traffic after EMD no longer has self-similarity. On this basis, IMFs can be forecasted using ARIMA to obtain a higher short-term forecasting accuracy with lower complexity.

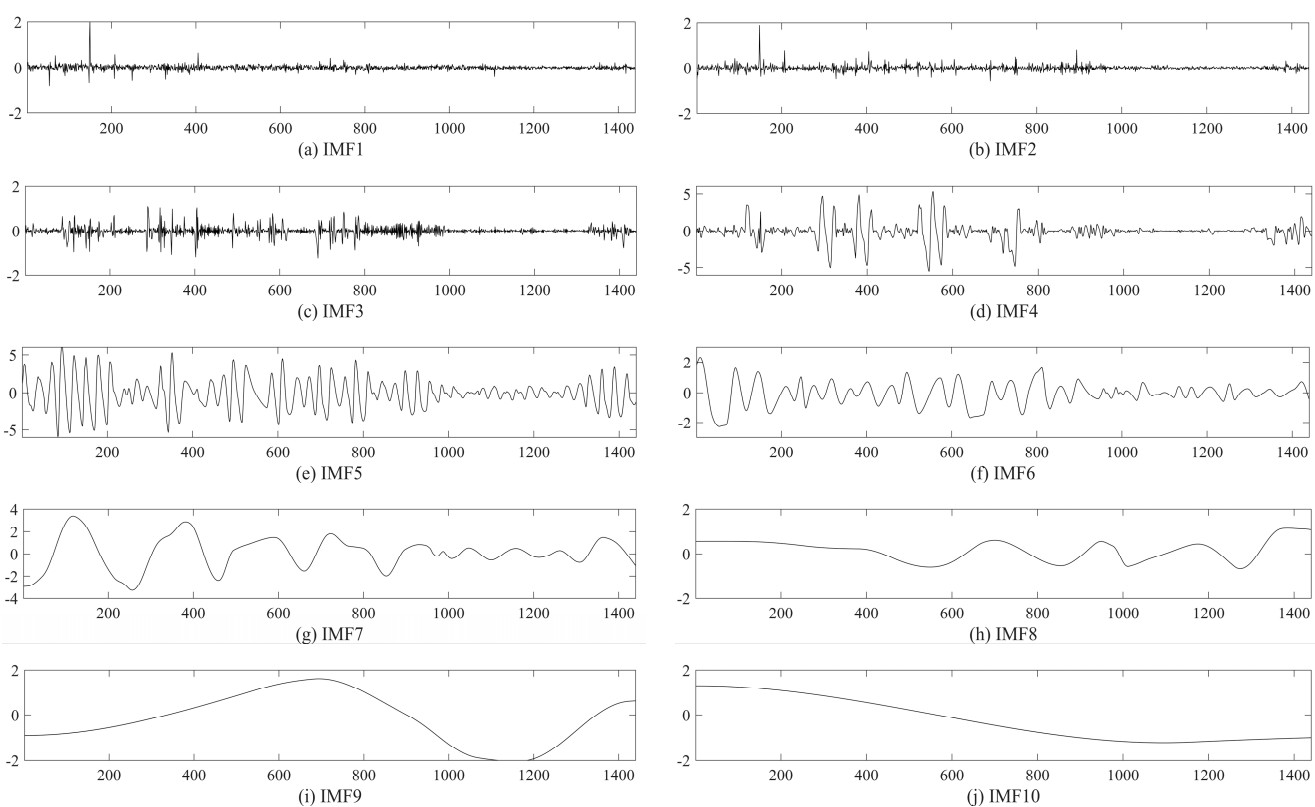

**Figure 10.** Intrinsic mode functions (IMFs) obtained from decomposition of raw traffic.

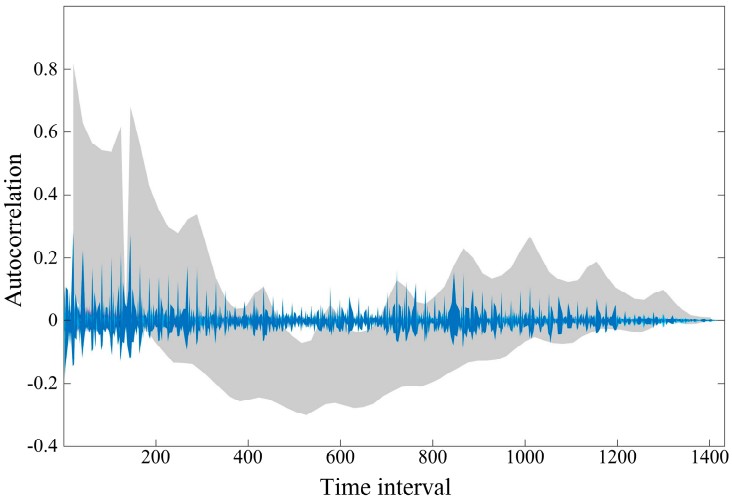

**Figure 11.** Autocorrelation function of IMF1~IMF10 vs. original data auto−correlation function.

*5.3. Satellite Network Traffic Forecasting*

After obtaining the IMFs and residuals of the satellite network traffic through EMD, ARIMA is utilized to forecast IMFs. The residuals are forecasted using the IGWO-BPNN model.

5.3.1. ARIMA-Based Forecasting for IMFs

This section follows the ARIMA modelling steps to construct ARIMA($p$, $d$, $q$) for each order of IMF. The optimal parameters of ARIMA($p$, $d$, $q$) for IMFs are determined via an adaptive fixed-order optimization search operator, as shown in Table 5.

**Table 5.** Optimal models corresponding to each order of IMFs.

| Parameters | $IMF_1$ | $IMF_2$ | $IMF_3$ | $IMF_4$ | $IMF_5$ | $IMF_6$ | $IMF_7$ | $IMF_8$ | $IMF_9$ | $IMF_{10}$ |
|---|---|---|---|---|---|---|---|---|---|---|
| $p$ | 1 | 2 | 2 | 2 | 2 | 2 | 2 | 2 | 2 | 2 |
| $d$ | 0 | 0 | 0 | 0 | 1 | 1 | 0 | 1 | 1 | 0 |
| $q$ | 1 | 0 | 0 | 0 | 0 | 0 | 0 | 0 | 0 | 0 |
| $AIC$ | $-12.9$ | $-15.4$ | $-18.5$ | $-22.4$ | $-32.3$ | $-37.4$ | $-33.9$ | $-47.6$ | $-54.4$ | $-67.2$ |
| $\phi_1$ | $-0.82$ | $1.72$ | $1.95$ | $1.99$ | $2.00$ | $2.00$ | $2.00$ | $2.00$ | $2.00$ | $2.00$ |
| $\phi_2$ | $0$ | $-0.88$ | $-0.99$ | $-1.00$ | $-1.00$ | $-1.00$ | $-1.00$ | $-1.00$ | $-1.00$ | $-1.00$ |
| $\theta_1$ | $-1.00$ | $0$ | $0$ | $0$ | $0$ | $0$ | $0$ | $0$ | $0$ | $0$ |
| $\sigma_\varepsilon^2$ | $2.4 \times 10^{-6}$ | $2.1 \times 10^{-7}$ | $9.0 \times 10^{-9}$ | $1.8 \times 10^{-10}$ | $9.7 \times 10^{-15}$ | $5.8 \times 10^{-17}$ | $1.8 \times 10^{-15}$ | $2.1 \times 10^{-21}$ | $2.3 \times 10^{-24}$ | $2.4 \times 10^{-25}$ |
| $\sigma_{IMF}$ | $1.6 \times 10^{-3}$ | $2.4 \times 10^{-3}$ | $3.1 \times 10^{-3}$ | $3.0 \times 10^{-3}$ | $2.0 \times 10^{-3}$ | $1.9 \times 10^{-3}$ | $1.7 \times 10^{-3}$ | $2.4 \times 10^{-3}$ | $2.4 \times 10^{-3}$ | $2.7 \times 10^{-3}$ |
| $\sigma_{IMF'}$ | $1.2 \times 10^{-5}$ | $5.3 \times 10^{-4}$ | $2.7 \times 10^{-3}$ | $2.8 \times 10^{-3}$ | $1.7 \times 10^{-3}$ | $1.9 \times 10^{-3}$ | $1.7 \times 10^{-3}$ | $2.4 \times 10^{-3}$ | $2.4 \times 10^{-3}$ | $1.9 \times 10^{-3}$ |
| $MAPE$ | $1.2 \times 10$ | $2.4 \times 10$ | $5.3 \times 10$ | $2.5 \times 10^{1}$ | $1.3 \times 10$ | $1.5 \times 10^{-1}$ | $5.1 \times 10^{-2}$ | $3.9 \times 10^{-3}$ | $1.3 \times 10^{-4}$ | $2.5 \times 10^{-4}$ |
| $RMSE$ | $1.6 \times 10^{-3}$ | $2.5 \times 10^{-3}$ | $3.4 \times 10^{-3}$ | $2.3 \times 10^{-3}$ | $5.2 \times 10^{-4}$ | $9.8 \times 10^{-5}$ | $1.3 \times 10^{-5}$ | $2.5 \times 10^{-6}$ | $9.2 \times 10^{-8}$ | $6.8 \times 10^{-8}$ |

In Table 5, $\phi_1, \phi_2, \theta_1$ and $\sigma_{\varepsilon,p,d,q}^2$ are the parameters of the EMD and $\sigma_{IMF}$ and $\sigma_{IMF'}$ are the standard deviations of the IMFs and fitted IMFs, respectively. The Akaike information criterion (AIC) is calculated as

$$AIC = T \cdot \ln\left(\sigma_{\varepsilon,p,d,q}^2\right) + 2(p + q + 1) \tag{26}$$

where $T$ denotes the length of the satellite network traffic. The mean absolute percentage error (MAPE) of the $i$-th IMF is calculated as

$$MAPE_i = \frac{1}{T}\sum_{t=1}^{T}\left|\frac{\xi_i(t)}{IMF_i(t)}\right| \tag{27}$$

The root mean square error (RMSE) of the $i$-th IMF is calculated as

$$RMSE_i = \sqrt{\frac{1}{T}\sum_{t=1}^{T}\xi_i(t)^2} \tag{28}$$

where $T$ denotes the length of the satellite network traffic and $\xi_i(t)$ denotes the $t$-th value of the fitted residual of $IMF_i$.

The fitted IMFs, i.e., $IMF_i'(t)$, are reconstructed to describe the linear feature of the satellite network traffic:

$$L'(t) = \sum_{i=1}^{n} IMF_i'(t) \tag{29}$$

5.3.2. IGWO-BPNN-Based Forecasting for Residuals

The IGWO-BPNN proposed in Section 4.2 is utilized to forecast residuals. The parameters of the IGWO and BPNN are shown in Table 6. Among the parameters of the BPNN, the number of neurons in the hidden layer directly affects the mapping ability of the BPNN to deal with complex problems. The empirical formula for defining the number of neurons is defined as

$$n = \sqrt{u + v} + b, \ b \in [1, 10] \tag{30}$$

where $u$ is the number of neurons in the input layer, $v$ is the number of neurons in the output layer and $b$ is the bias parameter.

The residual of the satellite network traffic is obtained by removing the linear sequence $L'(t)$ from the original sequence $D(t)$. In order to verify the forecasting performance of the IGWO-BPNN, the forecasting results $R'(t)$ for the IGWO-BPNN are compared with the results obtained from the BPNN, GWO-BPNN and grey-BPNN [35]. To ensure reasonable results, the parameters of the BPNN and GWO in the comparison algorithms are set to be the same as for the IGWO-BPNN.

**Table 6.** Initial parameters of IGWO-BPNN.

| Parameter | Value |
|---|---|
| Number of search agents of IGWO | 30 |
| Maximum iterations of IGWO | 500 |
| Number of neurons in input layer | 7 |
| Number of neurons in output layer | 1 |
| Number of neurons in hidden layer | 9 |
| Learning efficiency | 0.1 |
| Error limitation | $1 \times 10^{-3}$ |
| Maximum iterations of BPNN | 1000 |

The root mean square error (*RMSE*), mean absolute error (*MAE*), determination coefficient ($R^2$) and mean absolute percentage error (*MAPE*) are taken as the evaluation indexes of the forecasting results. The experimental results are shown in Figure 12 (Table 7 for specific values). Among them, the *MAPE* and *RMSE* are defined in Equations (26) and (27); *MAE* and $R^2$ are defined as

$$MAE = \frac{1}{T}\sum_{t=1}^{T}\left|R(t) - R'(t)\right| \tag{31}$$

$$R^2 = 1 - \frac{\sum_{t=1}^{T}\left(R(t) - R'(t)\right)^2}{\sum_{t=1}^{T}\left(R(t) - \overline{R}(t)\right)^2} \tag{32}$$

where $T$ denotes the length of the traffic data, $R(t)$ denotes the residual, $R'(t)$ denotes the forecasted residual and $\overline{R}(t)$ denotes the mean value of forecasted residual.

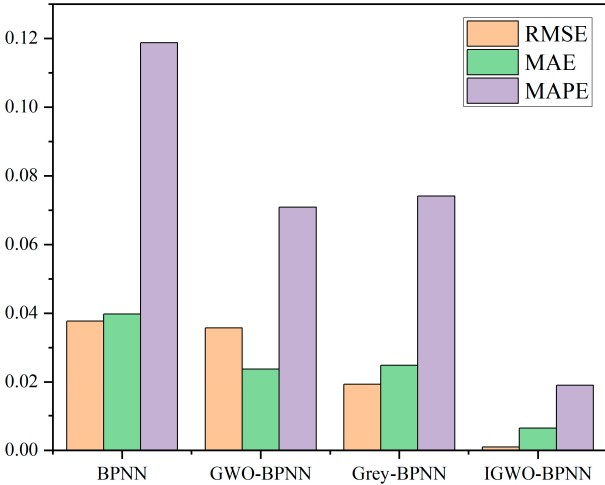

**Figure 12.** Forecasting error in residual series.

**Table 7.** Analysis of the forecasting performance of residual $R(t)$.

| Forecasting Schemes | Indexes | | | |
|---|---|---|---|---|
| | *RMSE* | *MAE* | $R^2$ | *MAPE* |
| BPNN | 0.037732 | 0.039750 | 0.87158 | 11.87% |
| GWO-BPNN [12] | 0.035736 | 0.023742 | 0.85732 | 7.09% |
| Grey-BPNN [35] | 0.019308 | 0.024802 | 0.92869 | 7.41% |
| IGWO-BPNN | 0.000935 | 0.006358 | 0.99735 | 1.90% |

As shown in Table 7, the IGWO-BPNN, GWO-BPNN and grey-BPNN represent a BPNN optimized using IGWO, GWO [12] and the grey model [35], respectively. Compared to the BPNN, IGWO-BPNN reduced the RMSE and MAE values by 97.4% and 84%, respectively, and improved $R^2$ by 14.43%. It is indicated that combining IGWO with BPNN is effective in forecasting the residuals of satellite network traffic.

### 5.3.3. Traffic Forecasting Based on the ARIMA-BPNN Hybrid Model

On the basis of obtaining the forecasted residual $R'(t)$, the forecasting result $D'(t)$ is obtained by integrating the linear sequence $L'(t)$ of traffic obtained from ARIMA($p_i, d_i, q_i$) according to Equation (24). The comparative analysis of the forecasted residuals proved the effectiveness of the IGWO algorithm to improve the BPNN. To verify the effectiveness of the proposed hybrid model on satellite network traffic forecasting, the forecasting results are compared with those from traditional forecasting methods, i.e., ARIMA, LSTM and ARIMA-BPNN. ARIMA-BPNN is a network traffic forecasting model proposed in [13]. The forecasting results of the above four methods are shown in Figure 13; the accuracy metrics and operation time are shown in Figure 14 (see Table 8 for specific values). The inputs of the EMD-ARIMA-BPNN, ARIMA, ARIMA-BPNN and LSTM are the testing dataset of the satellite network traffic.

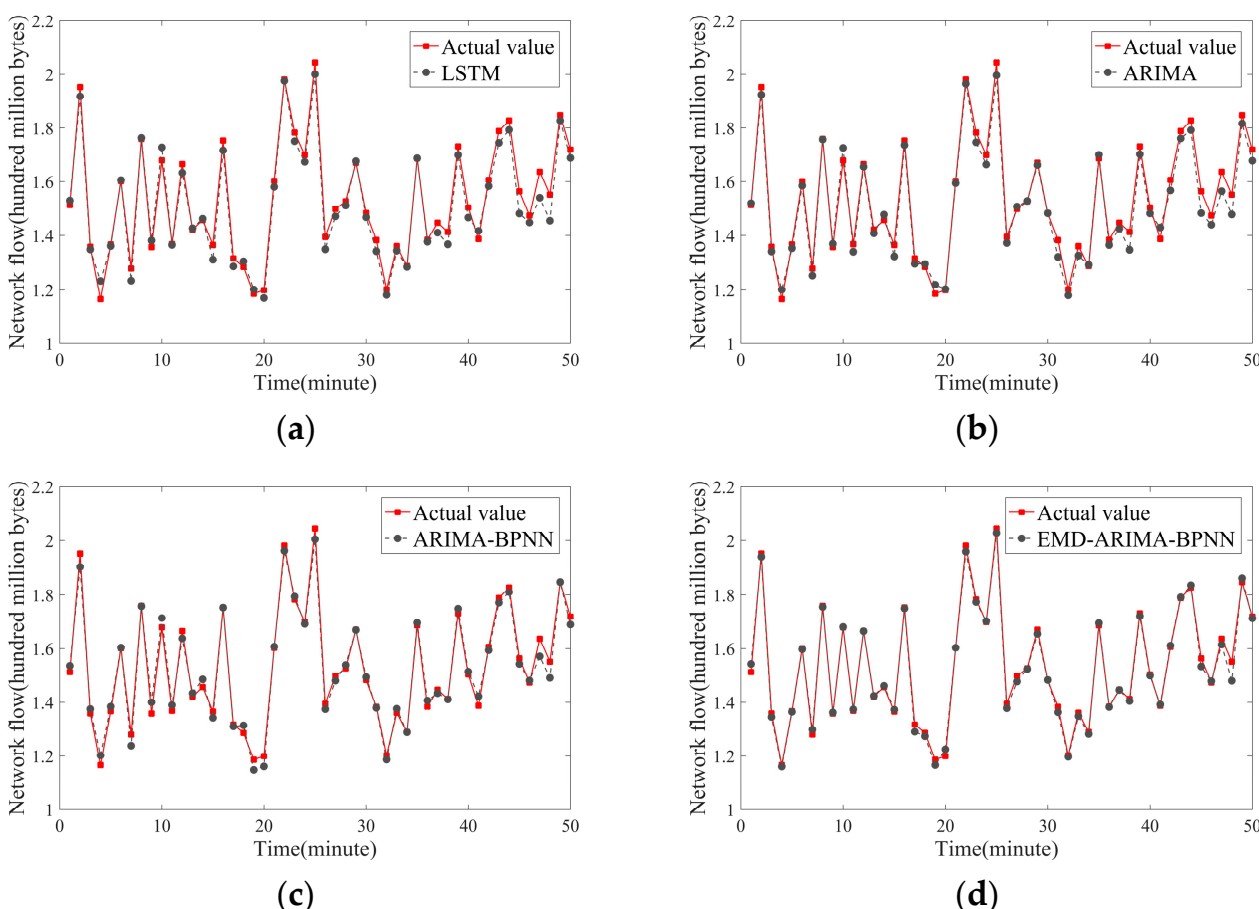

**Figure 13.** Satellite network traffic forecasting results. (**a**) Network traffic forecasted by LSTM. (**b**) Network traffic forecasted by ARIMA. (**c**) Network traffic forecasted by ARIMA-BPNN. (**d**) Network traffic forecasted by the proposed EMD-ARIMA-BPNN.

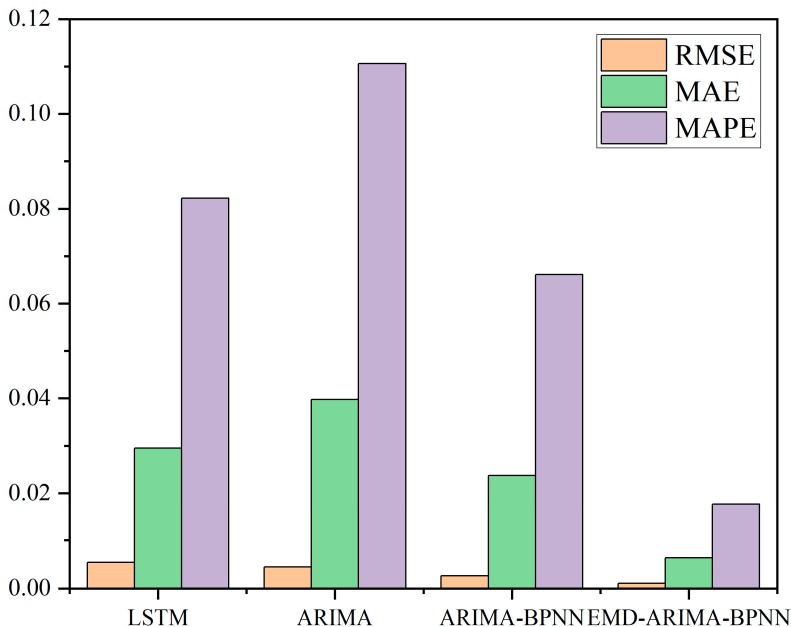

**Figure 14.** Forecasting error in satellite network traffic prediction.

**Table 8.** Analysis of the forecasting performance for satellite network traffic.

| Forecasting Schemes | Indexes | | | | |
|---|---|---|---|---|---|
| | *RMSE* | *MAE* | $R^2$ | *MAPE* | Time (s) |
| LSTM | 0.0054126 | 0.0294802 | 0.97899 | 8.22% | 4.159 |
| ARIMA | 0.0044703 | 0.0397500 | 0.99475 | 11.06% | 3.187 |
| ARIMA-BPNN | 0.0025979 | 0.0237420 | 0.98643 | 6.61% | 4.420 |
| EMD-ARIMA-BPNN | 0.0010770 | 0.0063580 | 0.98643 | 1.77% | 2.366 |

Combining Figure 13 and Table 8, it can be seen that the proposed hybrid model (EMD-ARIMA-BPNN) outperforms ARIMA and LSTM significantly on satellite network traffic with LRD characteristics. At the same time, EMD-ARIMA-BPNN also has advantages over ARIMA-BPNN in terms of error metrics and operation time. The forecasted traffic using EMD-ARIMA-BPNN is close to the actual value, which indicates the improved the forecasting accuracy for satellite network traffic.

5.3.4. Variable Step Forecasting

The above comparison experiments validate the forecasting performance of the proposed hybrid model for satellite network traffic. In order to verify whether the forecasting accuracy fluctuates significantly with the change in step size, the forecasting accuracy of the hybrid model is reanalyzed when varying step size [36]. The raw step size selected in this paper is measured in minutes. Therefore, *N*-step forecasting means that the satellite network traffic can be forecasted after *N* minutes. In particular, *N* = 1 indicates the forecasting target for the comparison experiments in the previous section. The variation in forecasting accuracy with different step sizes is shown in Figure 15.

The *MAPE* of EMD-ARIMA-BPNN tends to increase slowly as the forecasting step size increases. Therefore, EMD-ARIMA-BPNN can better describe the stage-by-stage trend of satellite network traffic, which provides an advantage for strategies such as the load-balanced routing of LEO mega constellations.

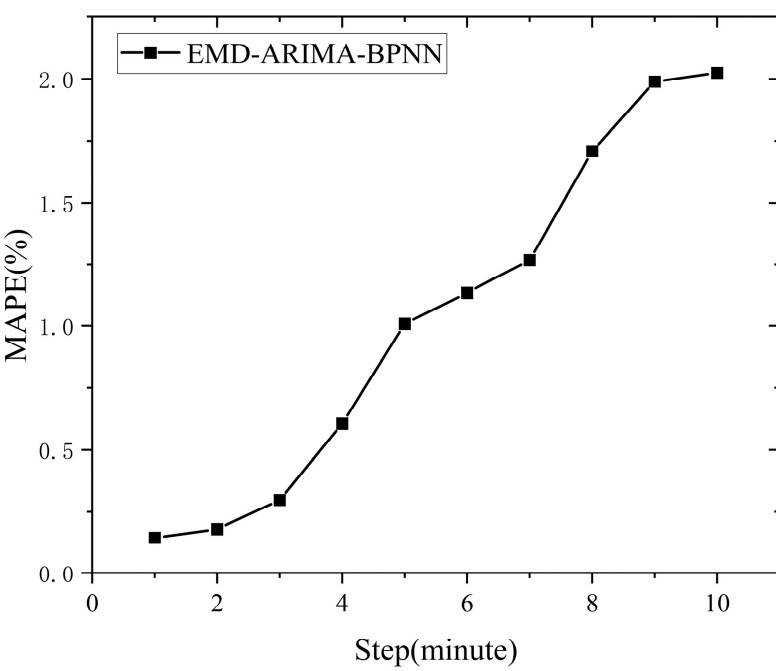

**Figure 15.** Variation in accuracy with different forecasting steps.

## 6. Conclusions

Satellite network traffic is characterized by self-similarity, long-range dependence and non-linearity, which makes it difficult to describe the traffic characteristics well with traditional single or hybrid models, resulting in low forecasting accuracy. This paper proposes an improved method for forecasting satellite network traffic by combining a BPNN with EMD-ARIMA. Different from traditional forecasting methods, we quantitatively analyze the self-similarity of satellite network traffic. Through theoretical analysis and experiments, the multi-order IMFs obtained from satellite network traffic via EMD have been proved to have short-range dependence. Thus, self-similar satellite network traffic is decomposed into multiple-order IMFs with short-range dependence using EMD. IMFs are forecasted by ARIMA and improved via an adaptive fixed-order search operator to reduce the computational complexity. At the same time, IGWO is adopted to optimize the hyperparameters of the BPNN. The residuals of EMD-ARIMA are forecasted using the optimized BPNN to finally obtain the forecasted satellite network traffic. Comparison experiments with traditional traffic forecasting models and hybrid models prove that the proposed EMD-ARIMA-BPNN model outperforms the traditional satellite network traffic forecasting methods, with several improvements in forecasting accuracy and efficiency.

In our future work, we will further apply the network traffic forecasting results to the routing of LEO mega-constellation networks. By sensing the change in traffic load, high-load ISLs can be avoided in advance, thus enhancing the load balancing ability of the mega-constellation network.

**Author Contributions:** Conceptualization, C.H.; methodology, C.H.; software, R.Y.; validation, W.X.; formal analysis, C.H.; investigation, W.X.; resources, W.X.; data curation, C.H.; writing—original draft preparation, C.H.; writing—review and editing, C.H.; visualization, C.H.; supervision, W.X.; project administration, W.X. and R.Y.; funding acquisition, W.X. and R.Y. All authors have read and agreed to the published version of the manuscript.

**Funding:** This research was funded by Space Engineering University grant number 614201003022207.

**Data Availability Statement:** The relevant data of this paper can be accessed by contacting hanchi@hgd.edu.cn.

**Conflicts of Interest:** The authors declare no conflicts of interest.

## Appendix A

**Proof of Lemma 1.** Let the Fourier transform of $C_i(t)$ in Equation (5) be $C_i(\omega)$ and the autocorrelation function be $R_c(\tau)$. According to the Fourier transform of the autocorrelation function and the power spectral density $S_X(\omega)$, we can obtain

$$R_c(\tau) = \frac{1}{2\pi} \int S_X(\omega) e^{j\omega\tau} \mathrm{d}\omega \tag{A1}$$

The signal power spectral density $S_X(\omega)$ is

$$S_X(\omega) = \lim_{T\to\infty} \frac{1}{T} |C_i(\omega)|^2 \tag{A2}$$

where $T$ is the length of the signal in the time domain and, according to IMF Definition 1, the IMF is similar to a smooth narrowband Gaussian process, so, $C_i(t)$ is a bandlimited signal. Substituting Equation (A2) into Equation (A1), we obtain

$$R_c(\tau) = \frac{1}{2\pi} \int \lim_{x\to\infty} \frac{1}{T} |C_i(\omega)|^2 e^{j\omega\tau} \mathrm{d}\omega = \frac{1}{2\pi T} \int |C_i(\omega)|^2 e^{j\omega\tau} \mathrm{d}\omega \tag{A3}$$

Since $R_c(\tau)$ is a real function, Equation (A3) can be simplified as

$$R_c(\tau) = \frac{1}{2\pi T} \int |C_i(\omega)|^2 \cos(\omega\tau) \mathrm{d}\omega \tag{A4}$$

Let the maximum and minimum values of $|C_i(\omega)|^2$ in the frequency band $\Omega$ be $M$ and $m$, respectively, then

$$\frac{m}{2\pi T} \int_\Omega \cos(\omega\tau) d\omega \leq R_c(\tau) \leq \frac{M}{2\pi T} \int_\Omega \cos(\omega\tau) d\omega \tag{A5}$$

i.e.,

$$R_c(\tau) \sim \int_\Omega \cos(\omega\tau) d\omega = \frac{1}{\tau} \sin(\omega\tau)|_\Omega \tag{A6}$$

Therefore, $R_c(\tau)$ is integrable. Thus, it is proven that the individual IMF components obtained from self-similar network traffic after EMD decomposition have SRD characteristics. □

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
