# Peer review of "A Hybrid Forecasting Model for Self-Similar Traffic in LEO Mega-Constellation Networks"

_aerospace, doi:10.3390/aerospace11030191_

Round 1

Reviewer 1 Report

Comments and Suggestions for Authors

For most part of the paper explanations are very good and helpful. Authors combined several existing approaches to tackle the prediction of traffic. However, few parts require clarification to avoid confusion and misunderstanding, therefore I have recommended major revision (but is more of a request for a sort of middle revision):

- Contribution listed in lines 93-97 should be written more coherently. The first sentence declares one contribution (ON-OFF model) but the following sentences do not seem connected to this sentence because they discuss the complexity and accuracy in a different manner unrelated to the first sentence.

- Figure 3 - what is considered as receiver, is it satellite node or ground station or something else, please clarify.

- Figure 5 should be accompanied by the explanation in the text. Same comment for the Algorithm 1. This should be done in manner used for other algorithms in the paper (for example, in lines 259-275)

- Text in lines 484-488 should be clarified. It is not clear what is trying to be said. This text does not seem to correlate to Figure 3. In Figure 3 sources are not in the satellite network, but here text says that satellites are sources, and then that the generated traffic is overlaid (I am not sure what is meant by this) to ground stations. I have understood it that finally ground stations send traffic to satellite network but then what is the purpose of initial satellite source nodes and overlaying process.

- If  I understood explanations and results correctly, results relate to joint traffic of all 8 pairs from Table 4. This does not completely make sense. Yes, overall traffic amount is important, but more important is where that traffic moves (originates and sinks) because theoretically it can be heavy overall traffic but all pairs are "direct" neighbors so none of the pairs affect other pairs. It would make more sense to predict separately traffic for each pair and use that information to optimize routing and resource allocation. Maybe, this is actually done but it is not clear from the text so this should be explained and clarified. Also, a brief discussion how the obtained prediction can be used might be added as for example, future work. For example, how the satellite dynamic moving from the ground station perspective affects the use of predictions to optimize traffic servicing by the satellite network point of view.

Comments on the Quality of English Language

I suggest proofreading to clear grammar and other errors in the text. Few examples:

line 82 - have (has)

line 228 - are (is)

line 79 - Parameters (parameters)

Reviewer 2 Report

Comments and Suggestions for Authors

In this paper, the authors have  proposed a mega-constellation network traffic forecasting model based

on EMD -ARIMA and IGWO optimized BPNN. Some of my comments are as follows:

1) The authors have mentioned that the proposed model outperform traditional constellations network traffic. Please discuss in details how this model outperform traditional constellations network.

2) The authors have utilized linear model ARIMA, non-linear model BPNN and optimization algorithm IGWO. What is the advantage of using these?

3) Please discuss the novelty of this work clearly.

4) Please rewrite the “Introduction” section so that it reflects the current state of arts.

5) Please discuss the limitations of this work and discuss the how it can be further improved.

Comments on the Quality of English Language

Moderate English editing is required.

Reviewer 3 Report

Comments and Suggestions for Authors

This paper presented a hybrid forecasting model for self-similar traffic in LEO  mega-constellation network. The paper was well-written with supportive evaluation results. The paper is recommended for publication as it is. The reviewer has only one minor comment. The motivation of the proposed approach is to reduce computation complexity of on-board LEO computer. Unfortunately, the reviewer could not find any results in the paper discussing how computation complexity had been reduced by the proposed approach. It would be great if the authors can elaborate this point more thorough. 

Round 2

Reviewer 1 Report

Comments and Suggestions for Authors

Almost all given suggestions are addressed properly, except the following:

"Figure 5 should be accompanied by the explanation in the text. Same comment for the Algorithm 1. This should be done in manner used for other algorithms in the paper (for example, in lines 259-275)."

The explanation needs to be added in the text as the figures themselves are not completely clear or informative. Also, placing figures/tables without proper addressing in text limits their value. The explanations can be fit in a reasonable space so the space limitation is not really issue. For example, in Figure 5 every block can be addressed with one or two proper sentences giving their purpose/goal and this can probably fit in 10 lines which in not such a big increase in text amount.

Reviewer 2 Report

Comments and Suggestions for Authors

Thanks for addressing my comments.

Comments on the Quality of English Language

Moderate English correction is required.
